# Effects of Baking and Frying on the Protein Oxidation of Wheat Dough

**DOI:** 10.3390/foods12244479

**Published:** 2023-12-14

**Authors:** Ru Liu, Yuhui Yang, Xiaojie Cui, Fred Mwabulili, Yanli Xie

**Affiliations:** 1College of Food Science and Engineering, Henan University of Technology, Zhengzhou 450001, China; lr0821ttkx@163.com (R.L.); yangyuhui1992@126.com (Y.Y.); 2021930535@stu.haut.edu.cn (X.C.); fred.mwabulili@stu.haut.edu.cn (F.M.); 2Henan Key Laboratory of Cereal and Oil Food Safety Inspection and Control, Henan University of Technology, Zhengzhou 450001, China

**Keywords:** baking, frying, protein oxidation, dough, oxidation products

## Abstract

Protein oxidation caused by food processing is harmful to human health. A large number of studies have focused on the effects of hot processing on protein oxidation of meat products. As an important protein source for human beings, the effects of hot processing on protein oxidation in flour products are also worthy of further study. This study investigated the influences on the protein oxidation of wheat dough under baking (0–30 min, 200 °C or 20 min, 80–230 °C) and frying (0–18 min, 180 °C or 10 min, 140–200 °C). With the increase in baking and frying time and temperature, we found that the color of the dough deepened, the secondary structure of the protein changed from α-helix to β-sheet and β-turn, the content of carbonyl and advanced glycation end products (AGEs) increased, and the content of free sulfhydryl (SH) and free amino groups decreased. Furthermore, baking and frying resulted in a decrease in some special amino acid components in the dough, and an increase in the content of amino acid oxidation products, dityrosine, kynurenine, and N’-formylkynurenine. Moreover, the nutritional value evaluation results showed that excessive baking and frying reduced the free radical scavenging rate and digestibility of the dough. These results suggest that frying and baking can cause protein oxidation in the dough, resulting in the accumulation of protein oxidation products and decreased nutritional value. Therefore, it is necessary to reduce excessive processing or take reasonable intervention measures to reduce the effects of thermal processing on protein oxidation of flour products.

## 1. Introduction

Protein oxidation occurs all the time during the process of food farm to table, especially in the process of processing, which changes the structure, sensory characteristics, and nutritional value of the protein. Reactive oxygen species (ROS) or oxidation products produced by foods under thermal processing directly or indirectly induce covalent modification of proteins to form protein oxidation/amino acid products [1,2], such as carbonyl [3,4], free sulfhydryl (SH) [5], dityrosine [6,7], and advanced glycation end products (AGEs) [8,9]. The intake of foods high in these oxidation products can lead to the accumulation of these substances in the body, which is closely related to obesity, diabetes, cardiovascular disease, and aging-related diseases (such as neurodegeneration, osteoporosis, and cataracts) [10]. Oxidized protein can induce damage to homeostasis through a variety of mechanisms, and may mistakenly bind to proteins such as enzymes and structural elements in cells, leading to dysfunction, apoptosis, and disease [7,11,12,13]. In particular, studies have found that protein carbonyl levels are elevated in patients with colorectal cancer, diabetes, neurodegenerative diseases, and aging, suggesting that protein oxidative damage is involved in the pathogenesis of these diseases [14,15]. It has been shown that long-term intake of dityrosine can induce oxidative stress and metabolic syndrome, disrupt thyroid hormone function, and increase the risk of obesity, diabetes, cardiovascular disease, learning, memory impairment, and accelerated aging [6]. AGEs as a highly complex and heterogeneous product of the reaction of reducing sugars with proteins and amino acids, a large number of studies have been reported to be associated with inflammation, chronic diseases, and cancer [16,17,18]. The degree of cross-linking and aggregation of protein is closely related to the nutritional value and bioavailability of the food itself. Protein oxidation induced by baking and frying may lead to cross-linking, aggregation, and fragmentation between proteins and amino acids [19,20], which will seriously reduce the digestion characteristics of proteins [21]. Undigested proteins enter the large intestine and are metabolized by intestinal microorganisms, which can produce toxic substances (hydrogen sulfide, ammonia, and phenol) and endanger human health [22].

Wheat is recognized as one of the three major cereals in the world, and its products are important food in many countries in the world. At present, the processing methods of flour products are various, and the commonly used processing methods are baking, frying, cooking, and so on. Van Steertegem et al. [23] showed the cross-linking of proteins in flour particles after heat treatment (for 2 or 5 h at 80 or 100 °C). A study pointed out that a cross-linking reaction occurs during the processing of wheat dough, resulting in disulfide bonds and dityrosine cross-linking products [24]. Bak et al. [25] used hot steam treatment of dough at 120 °C for 5, 10, and 15 min, and found that the physical properties such as hardness and texture of the dough changed. Zhou et al. [26] found that microwave heating could reduce the content of free SH groups in the dough. Kukurová et al. [27] measured the amino acid content of fried dough at 180 or 200 °C for 4, 6, and 8 min, and found that the content of aspartic acid and glutamic acid decreased significantly. At present, the research on the mechanism of thermal processing on protein oxidation mainly focuses on meat protein [28,29]. The research on flour products is not comprehensive, and its oxidation law is still unclear. Although the protein content of flour products is not high (10–15%), it is the main source of protein for residents in many countries with this staple food due to its large intake. Hence, it is of significance to clarify the impacts of baking and frying time and temperature on dough protein oxidation.

Therefore, under the assumption that baking and frying will lead to the oxidation of flour products, this study takes dough as the research object. Firstly, the oxidation of protein is evaluated by measuring the color, secondary structure, and oxidation products of protein (carbonyl, free SH, AGEs, free amino). Furthermore, the oxidation degree of amino acids was evaluated by amino acid composition analysis and marker amino acid oxidation products. In addition, the nutritional value of dough is evaluated by measuring the free radical scavenging rate and digestibility of dough.

## 2. Materials and Methods

### 2.1. Materials

Wheat flour (protein 12.2 g/100 g, dietary fiber 3.7 g/100 g, ash 0.58 g/100 g, fat 1.6 g/100 g, carbohydrate 72.3 g/100 g, and moisture 13.78 g/100 g) was obtained from Yihai Kerry Golden Dragonfish (Zhengzhou, China). Soybean oil (Yihai Kerry Golden Dragon fish) was purchased from Yonghui supermarket (Zhengzhou, China). The chemical reagents used in this experiment are analytically graded.

### 2.2. Samples Preparation

The dough (15 g, flour:water = 2:1, *m*:*v*) was prepared with the same length, width, and height. Subsequently, the doughs were separately heated as follows. The oven was preheated for half an hour. (A) Baking-Time: The dough was placed in the oven and baked at 200 °C for 0, 5, 10, 15, 20, 25, and 30 min, respectively. It was turned over once in half the time. (B) Baking-Temperature: The doughs were placed in the oven for 20 min and turned once at 10 min. The baking temperatures were 80, 110, 140, 170, 200, and 230 °C, respectively. The oil bath was preheated to the desired temperature. (C) Frying-Time: The doughs were fried at 180 °C for 0, 3, 6, 9, 12, 15, and 18 min. (D) Frying-Temperature: The temperature of the oil bath pot was stabilized at about 140, 150, 160, 170, 180, 190, and 200 °C, respectively. The dough was placed in the oil bath pot and fried for 10 min. The above preparation of samples was triplicated. The prepared samples were freeze-dried, ground, and stored at −20 °C. All samples were determined for protein content by the Kjeldah method.

The protein solution was extracted from the dough according to the method described by Qin et al. [5]. The obtained samples had fat removed using n-hexane. The defatted sample was dissolved in a phosphate-buffered solution (PH 7.0). The protein concentration in the supernatant was determined by Coomassie Brilliant Blue R250 (Jiancheng Bioengineering Institute, Nanjing, China). The sample was stored at 4 °C and used within 48 h.

### 2.3. Color Determination

The color of the dough was measured by Konica Minolta CR-400 (CR-A44, Konica Minolta Sensing Inc., Tokyo, Japan). Each sample was measured three times. The lightness (L*), redness (a*), and yellowness (b*) values of the samples were recorded.

### 2.4. Secondary Structure of Dough Protein

The secondary structure of the dough protein was determined via Fourier transform infrared (FTIR) spectroscopy (ALPHA, Bruker company, Karlsruhe, Germany). The 10 mg sample was added to 1 g KBr, and the flour-KBr mixture was added to an agate mortar, ground, and mixed in the light with the mold in the tablet after testing. The infrared spectrum was scanned by OPUS software (Version 7.5), and the test range was 4000~400 cm^−1^ with 16 scans with a resolution of 32^−1^.

### 2.5. Amino Acid Composition Analysis

The amino acid content of the sample was determined according to the method described by Zhou et al. [30] with some modifications. The sample (100 mg) was dissolved in 10 mL of 6 M HCl (containing 1 g/L phenol) filled with nitrogen for 2 min, and the lid was tightened under nitrogen conditions. The anaerobic tube was hydrolyzed in an oven at 110 °C for 22 h and filtered after cooling. The filtrate was evaporated 1~2 times by a rotary concentrator, and 1 mL diluent of the sample was added. The filter membrane was filtered and used for the determination of the automatic amino acid analyzer (Sykam S433D, Sykam GmbH, Munich, Germany).

### 2.6. Carbonyl Determination

The carbonyl content was measured according to the method described by Soglia et al. [31] with slight modifications. The 1.5 mL (10 mg/mL) sample solution was mixed with 5 mL of 10 mM 2,4-dinitrophenylhydrazine (DNPH, dissolved in 2 M HCl) and the mixture was incubated for 1 h at room temperature under darkness. The 1.5 mL sample solution was mixed with 5 mL 2 M HCl as a control. After incubation, 0.5 mL of trichloroacetic acid (20%) was added and shaken. After centrifugation (10,000× *g*, 4 °C, 15 min), the precipitate was washed 3 times with 1 mL of C_2_H_5_OH-C_4_H_8_O_2_ (1:1) mixed solution. Next, the mixture was dissolved in 1.5 mL of guanidine hydrochloride (PH 7.0), reacted for 1 h at 25 °C under darkness, and colorimetric at 367 nm (UV-6100S, MAPADA company, Shanghai, China).

### 2.7. Free SH Determination

The free SH content was measured according to the method described by Zhou et al. [32] with a slight modification. A total of 150 mg sample was added with 5 mL buffer solution (10 mM ethylenediamine tetraacetic acid, 0.2 M glycine, 20 mM Tris) and 50 μL of 4 mM 5,5′-Dithiobis-(2-nitrobenzoic acid) (DTNB) solution. The supernatant was taken for colorimetric analysis at 412 nm. The free SH content was determined according to the following formula: SH (nmol/g) = 73.53 × A_412_/C; A367 represents the absorbance of free SH at 412 nm; C represents the protein concentration.

### 2.8. Free Amino Content Determination

The free amino content was assessed according to the method described by Gujral and Rosell [33] with some modifications. The absorbance of 200 μL dough protein sample and 1.5 mL buffer solution (3.81 g sodium tetraborate, 0.1 g SDS; 80 mg o-Phthalaldehyde was dissolved in 2 mL ethanol, the mixed solution was dissolved in 100 mL water with 88 mg Dithiothreitol) in dark for 3 min was measured at 340 nm. The standard curve of the free amino group was obtained by using 0–200 mg/L L-serine, and the content of the free amino group in protein was further calculated.

### 2.9. AGEs Content Determination

The AGEs content was assessed according to the method described by Wang et al. [34] with a slight modification. An amount of 250 mg of sample was mixed with 5 mL buffer solution (Tween 20: sodium dodecyl sulfate: mercaptoethanol, 0.05%:1%:5%, *v:v:v*, PH 7.4, 50 mmol/L Tris-HCl), and shaken in a water bath at 37 °C for 5 h and then transferred to 37 °C constant temperature shaker for 6 h. The fluorescence intensity of the supernatant was measured. The excitation wavelength (Ex) of fluorescence detection was 390 nm and the emission wavelength (Em) was 480 nm (F7100, HITACHI Corporation, Tokyo, Japan).

### 2.10. Dityrosine Cozntent Determzination

The dityrosine was assessed according to the method described by Xiong et al. [35] with a minor modification. The 200 mg sample was accurately weighed in the hydrolysis tube, slowly added with 8 mL of 6 M HCl, hydrolyzed in the oven (110 °C, 24 h), cooled, and diluted to a 100 mL volumetric flask. The supernatant was taken as the test solution. The Ex was 292 nm and the Em was 410 nm determined via fluorescence spectroscopy.

### 2.11. Tryptophan Determination

The tryptophan content was evaluated by Estévez et al. [36] with a slight modification. A total of 250 mg of the samples was dissolved in 5 mL phosphate buffer (PH 7). Centrifugal (8000× *g*, 5 min) was performed after a water bath (3 h, 37 °C). Fluorescence values were measured under Ex 283 nm, Em 250–400 nm, and emission slit widths of 5 nm, and the scanning speed was 1000 nm/min.

### 2.12. Determination of Kynurenine and N’-Formylkynurenine

The content of kynurenine and N’-formylkynurenine was assessed according to the method described by Wang et al. [34] with slight modifications. The sample treatment method is the same as the determination of AGEs. The fluorescents were measured by Ex/Em 365/480 nm and 325/434 nm.

### 2.13. Determination of Free Radical Scavenging Ability

The determination of free radical scavenging ability (ABTS^+^, DPPH) was assessed by Rizzello et al. [37] with a minor modification.

ABTS^+^: The 0.6 mL sample solution (10 mg/mL) was mixed with 2.4 mL ABTS^+^ solution (7 mM ABTS: 2.45 mM K_2_S_2_O_8_, 1:1, *v:v*, A_734_ = 0.70 ± 0.02). The absorbance was measured at 734 nm. ABTS^+^ scavenging ability (%) = (A_0_ − A)/A_0_ × 100%; A_0_ represents the absorbance value of 2.4 mL ABTS^+^ working solution after reaction with 0.6 mL anhydrous ethanol. A represents the absorbance value of 2.4 mL ABTS^+^ working solution after reacting with 0.6 mL sample solution.

DPPH: Briefly, 2 mL of sample solution (10 mg/mL) was mixed with 2 mL of 0.2 mmol/L DPPH solution and kept in the dark to react for 30 min, and the absorbance was measured at 517 nm. DPPH scavenging ability (%) = (1 − (A_i_ − A_j_)/A_c_) × 100%; A_i_ represents the absorbance of the 2 mL sample after reacting with 2 mL 0.2 mmol/L DPPH solution. A_j_ represents the absorbance of the 2 mL sample after reacting with 2 mL absolute alcohol. A_c_ represents the absorbance of 2 mL absolute alcohol after reacting with 2 mL of 0.2 mmol/L DPPH solution.

### 2.14. In Vitro Protein Digestibility (IVPD)

IVPD was determined as described by Liu and Zhao [38] with a minor modification. The IVPD was determined by gastric-trypsin two-step digestion. The sample (500 mg) was mixed with the reaction mixture (1.5 mg of pepsin was added to 15 mL of 1.5 N HCl) and digested in a shaken water bath at 37 °C for 3 h, and neutralized with 0.2 N NaOH to pH 8. Then, it was digested in 7.5 mL of phosphate buffer containing 4 mg pancreatin in a shaken water bath at 37 °C for 2 h, and 5 mL of trichloroacetic acid was added to terminate the reaction. After centrifugation (5000× *g*, 20 min), the supernatant was taken to determine the free radical scavenging rate. The protein content of the precipitated samples was determined by the Kjeldahl method after freeze-drying. IVPD was determined by using the following formula: IVPD (%) = (M_0_ − M_1_)/M_0_ × 100%; M_0_ represents the protein content of dough samples before digestion; M_1_ represents the protein content of dough samples after digestion.

### 2.15. Principal Component Analysis (PCA)

PCA uses SIMCA-P+ software (version 14.1, Umetrics, Umea, Sweden) for multivariate data analysis. Based on the above 20 groups of data (L*, a*, b*, α-helix, random coil, β-sheet, β-turn, carbonyl, free SH, AGEs, free amino, tryptophan, and its oxidation products, dityrosine, ABTS^+^, DPPH, digestibility, etc.) as the object, the difference analysis of different baking and frying time and temperature groups was carried out. The calculation parameters of R^2^ (cum) and Q^2^ (cum) are used to evaluate PCA. R^2^X > 0.50 and Q^2^ > 0.50 indicate that the model is robust and has good fitness and prediction [39].

### 2.16. Statistical Analysis

All samples and tests were replicated 3 times. All experiment data were analyzed using SPSS Statistics 23, and the results were expressed as mean ± standard error mean (SEM). The one-way ANOVA and Tukey’s HSD were used to analyze the differences between groups, and the results were expressed in different letters (*p* < 0.05). The independent sample *t*-test was used to analyze the differences between the data of each experimental group and the control group, and the results were expressed in “*” (*p* < 0.05) and “**” (*p* < 0.01). And Z-score = (the content of each sample-average content of all samples)/(standard deviation of value in all samples) [39].

## 3. Results

### 3.1. Baking and Frying Deepen the Dough Color

The color change in thermally processed food is generally used to directly evaluate the degree of protein oxidation, which is usually affected by the heating method, temperature, and time [40,41]. Figure 1 shows that the color of the sample gradually deepens with the increase in baking and frying time and temperature. The L*, a*, and b* values were used to reflect the color change in the samples [42]. Among them, as the baking and frying time and temperature are prolonged, the L* value significantly decreased (*p* < 0.05), while the a* and b* values significantly increased (*p* < 0.05). This phenomenon may be related to the glycosylation reaction between protein and carbohydrates in gluten [5]. The size of the b* value is directly related to the Maillard reaction, which produce some colored substances turning the food yellow [43]. Ouyang et al. [44] revealed that the color deepening of wheat flour was caused by protein oxidation, and the color difference was relevant to the protein oxidation index and the concentration of oxidation products. Li et al. [45] treated the dough with ozone and found that the color of the dough was deepened and the secondary structure was changed. Therefore, our above results indicate that baking and frying cause the color of the dough to deepen, presumably leading to changes in protein structure and oxidation.

### 3.2. Baking and Frying Changed the Secondary Structure of Dough Protein

To clarify the effects of baking and frying on the structure of dough protein, we analyzed the changes in the secondary structure of dough protein during baking and frying. The amide I band (1600–1700 cm^−1^) was subjected to baseline correction, Gaussian smoothing, and recoiling [46]. According to the ratio of each characteristic peak to the total peak area, the content of the secondary structure of the protein was obtained. The results are shown in Figure 2, which show that during baking and frying, α-helix and random coil had an obvious decreasing trend (*p* < 0.05); β-sheet had a significant upward trend (*p* < 0.05); the β-turn had an increasing trend in the baking treatment, but the increasing trend was not obvious in the frying treatment (*p* > 0.05). The overall performance was irregular curl, α-helix toward β-sheet, and β-turn direction change. Yin et al. [47] proved that hydrogen peroxide-induced protein oxidation significantly increased the proportion of β-sheet and β-turn, consistent with our results. The secondary structure of proteins may be affected by various forces such as covalent cross-linking, electrostatic interaction, and hydrophobic interaction. The reason for this change may be that heat treatment destroys the hydrogen bond of the α-helix inside the polypeptide chain, uncoils, and transforms into β-sheets under the interaction force [48]. The polypeptide chain will flip 180° under heat treatment, increasing β-turn [49]. Zhang et al. [50] proved that with the increase in the protein oxidation degree, the content of α-helix decreased, the content of β-sheet increased first and then decreased, and the content of β-turn decreased, which is consistent with our findings. Therefore, our results indicate that baking and frying cause changes in protein structure, which may lead to protein oxidation in the dough.

### 3.3. Baking and Frying Caused Cross-Linking and Oxidation of Dough Proteins

To further determine the changes in protein structure, we measured the free SH and AGE content to reflect the level of protein cross-linking. The free SH groups of amino acids are easily attacked by ROS and free radicals, and oxidation occurs to form disulfide bonds. Therefore, cross-linking can be characterized by the loss of SH groups. AGEs are a series of complex covalent adducts formed by reducing sugars, proteins, and amino acids, which are the landmark indicators of protein cross-linking. In Figure 3, as the baking and frying time and temperature prolonged, the content of free SH was decreased by 57% (A), 60% (B), 22% (C), and 16% (D), respectively (*p* < 0.05), and the fluorescence intensity of AGEs increased by 492% (A), 435% (B), 442% (C), and 471% (D), respectively (*p* < 0.05). The decrease in free SH may be because sulfur-containing amino acids are vulnerable to free radicals and cross-linked to form disulfide bonds [51,52]. The occurrence of this cross-linking phenomenon cause the protein to form insoluble aggregates, resulting in oxidative aggregation of proteins. However, during the frying process, the free SH content showed two stages of change, and the difference was statistically significant (*p* < 0.05). This may be because the frying process hinders the conversion of free SH groups to disulfide bonds. It has shown that the AGE contents are significantly positively correlated with the b* value [5] during the heat treatment. It suggested that the b* value can be used to evaluate the AGE levels visually, and the formation of AGEs is closely related to the lard reaction. High temperature and low moisture promote the lard reaction, thereby promoting the production of AGEs; some oxidizing substances can also be used as a bridge to connect the lard reaction and promote the formation of AGEs [53,54]. The loss of free SH groups and the increase in AGE macromolecules showed that the protein had different degrees of cross-linking aggregation during frying and baking, indicating that the dough protein was oxidized during baking and frying.

To more visually clarify the level of protein oxidation, we measured the content of carbonyl and free amino groups. Carbonyl is a product of protein oxidation, which is often used to evaluate the degree of protein oxidation. Protein oxidation will cause aggregation of free amino groups, and the content of free amino groups will decrease. As presented in Figure 3, as the baking and frying time and temperature prolonged, the content of carbonyl was increased by 165% (A), 161% (B), 239% (C), and 199% (D), respectively (*p* < 0.05), and the content of free amino groups decreased by 50% (A), 51% (B), 67% (C), and 63% (D), respectively (*p* < 0.05). Our results suggest that protein oxidation and accumulation of oxidation products are induced by increasing baking and frying time and temperature. In this study, baking and frying will aggravate the accumulation of carbonyl compounds, this phenomenon may be because high temperature will cause the formation and cracking of hydrogen peroxide and promote the production of oxidation reaction, increasing carbonyl content [55]. The oxidation of vulnerable amino acid residues or the oxidative cleavage of the protein backbones can produce carbonyl groups [56]. When the baking reached 5 min at 200 °C, 80 °C at 20 min, and the frying reached 3 min at 180 °C, 140 °C at 10 min had the highest free amino content. This may be because moderate protein oxidation induced the exposure of buried hydrophobic amino acids [57]. The side chain carbonylation of protein amino acids is one of the reasons for the decrease in free amino content [10]. This is consistent with the carbonyl content we measured. The amino acids with NH- and NH_2_- on the side chain of amino acids are very sensitive and vulnerable to ROS attacks. Wang et al. [58] used the hydroxyl radical model to induce egg white protein, and the free amino group decreased to varying degrees, which was consistent with our research results. The above results showed that the degree of protein oxidation increased with the extension of baking and frying time and temperature.

### 3.4. Baking and Frying Caused Amino Acid Oxidation in the Dough

In addition to the cross-linking phenomenon and the change of protein secondary structure, the oxidation of food proteins is also accompanied by the oxidation and cross-linking of amino acids. Thermal processing can easily cause amino acids to be attacked and oxidized, so the changes in amino acid content can directly reflect the degree of oxidation of amino acids. Figure 4 and Figure 5 show the changes in 18 amino acids in dough protein. Compared to the control group, the amino acid content generally decreased after baking and frying. As the baking and frying time and temperature prolonged, the quantity of Tyr, Cys, Met, Asp, Phe, His, Lys, Arg, and Pro was significantly less (*p* < 0.05), which showed that these amino acids were the main oxidation sites of the protein during baking and frying, and implied that amino acids are oxidized. In addition, by scanning the characteristic fluorescence spectrum of tryptophan (the fluorescence intensity of tryptophan was positively correlated with the concentration of tryptophan [10]), we found that the content of tryptophan was also significantly reduced, indicating that tryptophan was oxidized during baking and frying, and was also the main amino acid oxide site. Quansa et al. [59] quantitatively analyzed the amino acid content of Cowpa protein by heat treatment and found that thermal processing will lead to a decrease in the content of some sulfur-containing amino acids. This is consistent with our findings. Studies have found that amino acids are oxidized, mainly cross-linking reactions or oxidized to other compounds [60,61]. The decrease in tyrosine content is largely due to the cross-linking reaction. It has been reported that the wavelength values (around 350–360 nm) correspond to tryptophan residues mainly located on the surface of the protein, which proves that the heat treatment mainly acts on the tryptophan residues on the surface of the protein [36]. In the later baking and frying periods, the fluorescence intensity is not much different, this may be due to the dehydration of the dough sample caused by high temperature, the aggregation of all processed proteins is extensive, and the steric hindrance increases, which subsequently quenches the fluorescence intensity [62]. It is speculated that thermal processing as an initiation mechanism converts some amino acids into free radicals, which can immediately react with molecular oxygen and lipids to produce peroxide groups and hydrogen peroxide, and finally produce amino acid oxidation products [63].

The oxidation of amino acids result in the accumulation of amino acid oxidation products in food. To further clarify the effects of baking and frying on the oxidation of amino acids, representative amino acid oxidation products were detected. Dityrosine, as the oxidation product of tyrosine, is widely detected in food and has been considered a marker of protein oxidation. Kynurenine and N’-formylkynurenine as an intermediate of tryptophan catabolism are usually used to characterize the degree of amino acid oxidation. As presented in Figure 6, compared to the control group, the fluorescence intensity of dityrosine increased by 4.38 (A), 3.86 (B), 3.80 (C), and 4.16 times (D), respectively (*p* < 0.05); the fluorescence intensity of kynurenine increased by 3.69 (A), 3.31 (B), 3.33 (C), and 3.61 times (D), respectively (*p* < 0.05); and the fluorescence intensity of N’-formylkynurenine increased by 8.89 (A), 6.14 (B), 4.40 (C), and 5.41 times (D), respectively (*p* < 0.05) with the baking and frying time and temperature prolonged. Dityrosine is generated by the isomerization of two tyrosine radicals and finally enolization, and dityrosine exists in the form of covalent and noncovalent aggregates [64]. Dityrosine can be produced in large quantities under hot processing conditions such as high temperature, high pressure, and spray drying. Chen et al. [65] quantitatively determined the oxidation products of tyrosine and tryptophan in infant milk powder after different degrees of oxidative modification and found that dityrosine, kynurenine and N’-formylkynurenine increased to varying degrees, which is consistent with our results.

### 3.5. Baking and Frying Changed the Free Radical Scavenging Rate and Protein Digestibility

The ability of food to scavenge free radicals can indirectly reflect its antioxidant ability, which can be used as an indirect index to evaluate protein oxidation [21]. To clarify the effects of baking and frying on the free radical scavenging of dough, the ability of the dough to scavenge ABTS^+^ and DPPH was detected. As shown in Figure 7, we found that, compared with the control group, the free radical scavenging ability of ABTS^+^ decreased by 41% (A), 45% (B), 20% (C), and 54% (D), respectively (*p* < 0.05), and the free radical scavenging ability of DPPH decreased by 22% (A), 25% (B), 7% (D), and 39% (C), respectively (*p* < 0.05), indicating that frying and baking reduce the free radical scavenging rate, which also supports the conclusion that baking and frying cause dough protein oxidation. A study has shown that the antioxidative activity of egg white protein is significantly reduced under high-temperature environments, which also supports our conclusion [66]. Furthermore, the presence of free amino acids (tyrosine, tryptophan, and phenylalanine), methionine, cysteine, histidine, and lysine can improve the free radical quenching activity [67,68], thereby improving the antioxidant capacity of the oxidation products. However, the occurrence of oxidative modification of free amino acids during baking and frying will seriously affect the free radical antioxidant activity of oxidation products, which is consistent with our research results.

The digestibility of protein is usually used to evaluate the nutritional value of proteins, which is related to the degree of protein oxidation. As shown in Figure 8, when the Baking Time was 5–15 min, the Baking Temperature was 80–140 °C, the Frying Time was 3–6 min, and the Frying Temperature was 140–150 °C, the digestibility was obviously (*p* < 0.05) higher than that of the control group. After that, as baking and frying time and temperature prolonged, the digestibility was obviously (*p* < 0.05) lower than that of the control group. These results indicated that proper processing promotes protein digestion, while excessive processing reduces protein digestibility. Proteins are mainly digested by small molecular peptides and amino acids in the body. During digestion, pepsin decomposes proteins into small molecular weight amino acid active peptide, which has more efficient antioxidant characteristics and health benefits [69]. Ma et al. [70] also confirmed that protein carbonylation and cross-linking reduced the digestibility of protein. In our study, with the increased baking and frying time and temperature, the decrease in protein digestibility may be caused by protein oxidation. It has been shown that the recognition sites of pepsin are mainly hydrophobic amino acids [47], but with the oxidation of protein and the aggregation of macromolecular substances, the enzyme cannot recognize the hydrolysis point. A study has shown that linoleic acid induces soybean protein oxidation, and it was found that under low oxidation conditions, the digestibility increased, while under high oxidation conditions, the digestibility decreased [30], which is consistent with our conclusion. After digestion by protease, the free radical scavenging rate also presented a trend of increasing first and then decreasing, which was consistent with the trend of protein digestibility. It is speculated that the increase in free radical scavenging rate may be due to the hydrolysis of protein into more free amino groups and peptides in the early stage of digestion, which bind to free radicals. The decrease in free radical scavenging rate may be due to the formation of protein oxidation products and the decrease in free amino acids and peptides in the later stage of digestion. Recently, as people pay more and more attention to nutritional diets, it is urgent to evaluate the nutritional value and potential health threats of oxidized dietary proteins, thus calling for people to improve overprocessing and inhibit protein oxidation.

### 3.6. Principal Component Analysis (PCA)

PCA is an unsupervised pattern recognition method used to identify any trends or outliers in the data. Using the above 20 sets of data as principal components, different baking and frying times and temperatures were grouped to obtain Figure 9. From these four models, we can obtain some information. First, R^2^X and Q^2^ are more than 0.5, indicating that the model has good robustness, good fitness, and predictive ability. The scattered points are all distributed in the confidence interval, and there is no abnormal value, indicating that the data have certain statistical significance. The scatter points in each group showed a certain aggregation phenomenon, which proved that the sample had good repeatability. The scatter points between the different groups are more dispersed, indicating that there are some differences between the groups. In particular, the difference between the treatment groups and the control group was particularly obvious, indicating that the effects of baking and frying on protein oxidation were very significant. With the increase in baking and frying times and temperatures, there was a certain regularity in the scatter distribution between the different treatment groups, indicating that the effects of baking and frying time and temperature on protein oxidation were significant.

## 4. Conclusions

This study confirmed that with the increase in baking and frying time and temperature, the color of dough deepened and the secondary structure of protein changed. In addition, protein cross-linking occurs, protein oxidation products accumulate, and more importantly, part of the amino acid content decreases, and its oxidation product content increases. Moreover, free radical scavenging rate and digestibility have a downward trend. The above results show that baking and frying can lead to the oxidation of protein and amino acids and the degree of oxidation increases with the increase in processing time and temperature. This article calls on us to reduce excessive processing, especially long-time and high-temperature frying and baking. We need to further explore the methods to inhibit the oxidation of proteins in the process of food thermal processing and provide the scientific theoretical basis for the production of healthy and safe food.

## Figures and Tables

**Figure 1 foods-12-04479-f001:**
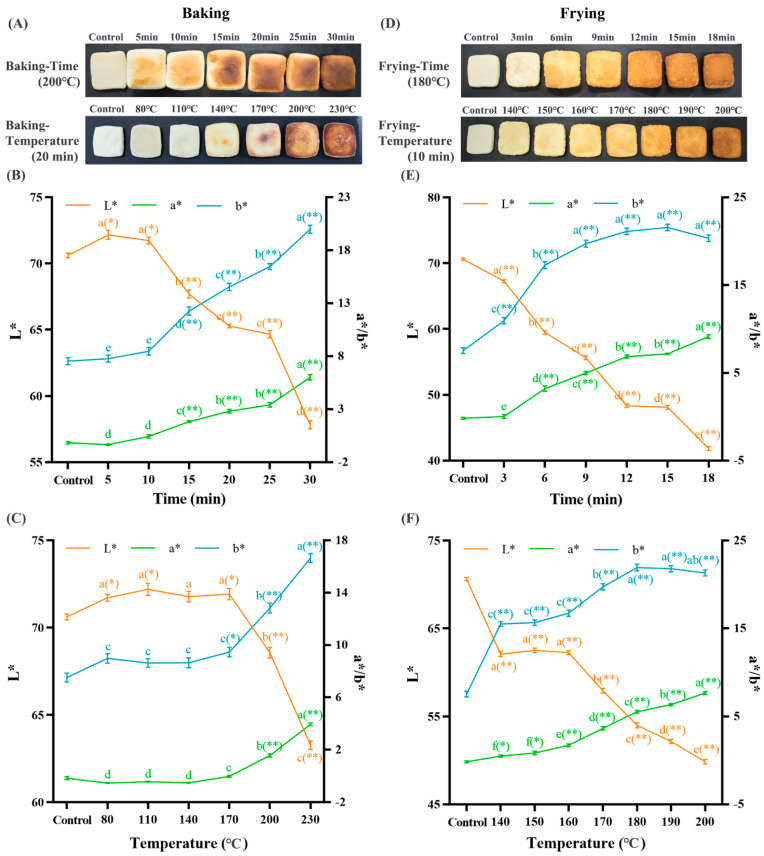
Effects of baking (**A**) and frying (**D**) on the overall appearance of the dough. Effects of baking time ((**B**) under 200 °C) and temperature ((**C**) under 20 min) on the color parameters of the dough. Effects of frying time ((**E**) under 180 °C) and temperature ((**F**) under 10 min) on the color parameters of the dough. The data are displayed as mean ± SEM (*n* = 3). Different lowercase letters indicate significant differences between different baking and frying times and temperatures; the markers “*” and “**” represente the significant differences (*p* < 0.05) and highly significant differences (*p* < 0.01) vs. the control group.

**Figure 2 foods-12-04479-f002:**
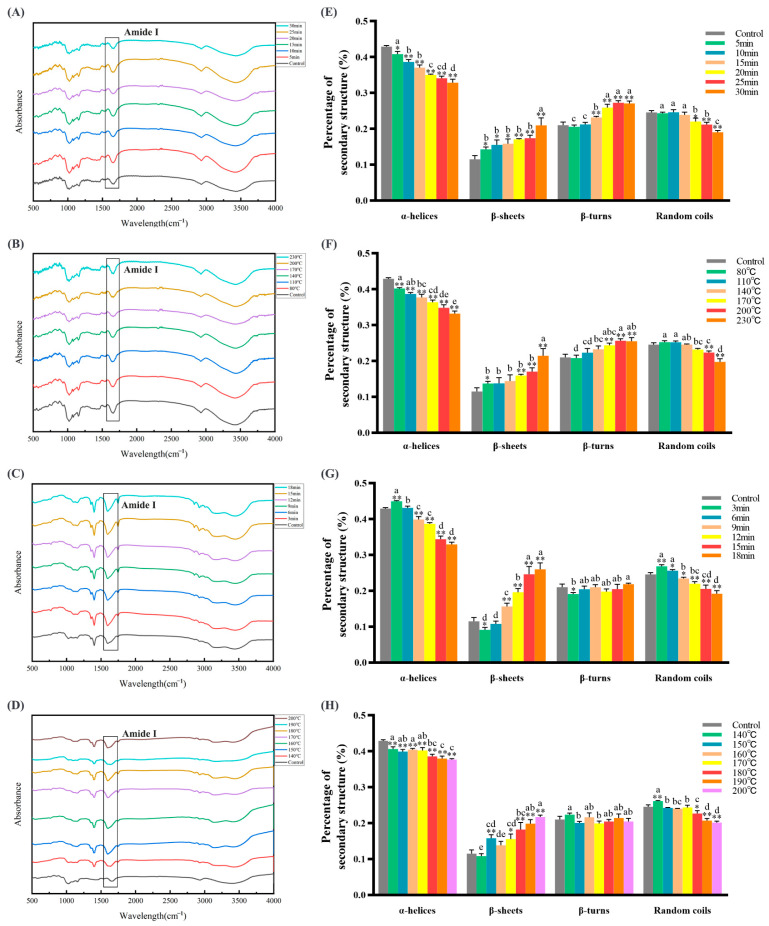
Full-wavelength scanning of fluorescence spectra at baking temperatures ((**A**) under 200 °C) and times ((**B**) under 20 min); Full-wavelength scanning of fluorescence spectra at frying temperatures ((**C**) under 180 °C) and times ((**D**) under 10 min); effects of baking times ((**E**) under 200 °C) and temperatures ((**F**) under 20 min) on the secondary structure of the dough protein; effects of frying times ((**G**) under 180 °C) and temperatures ((**H**) under 10 min) on the secondary structure of dough protein. Amide Ⅰ: 1600–1700 cm^−1^. The data are displayed as mean ± SEM (*n* = 3). Different lowercase letters indicate significant differences between different baking and frying times and temperatures; the markers “*” and “**” represent the significant differences (*p* < 0.05) and highly significant differences (*p* < 0.01) vs. the control group.

**Figure 3 foods-12-04479-f003:**
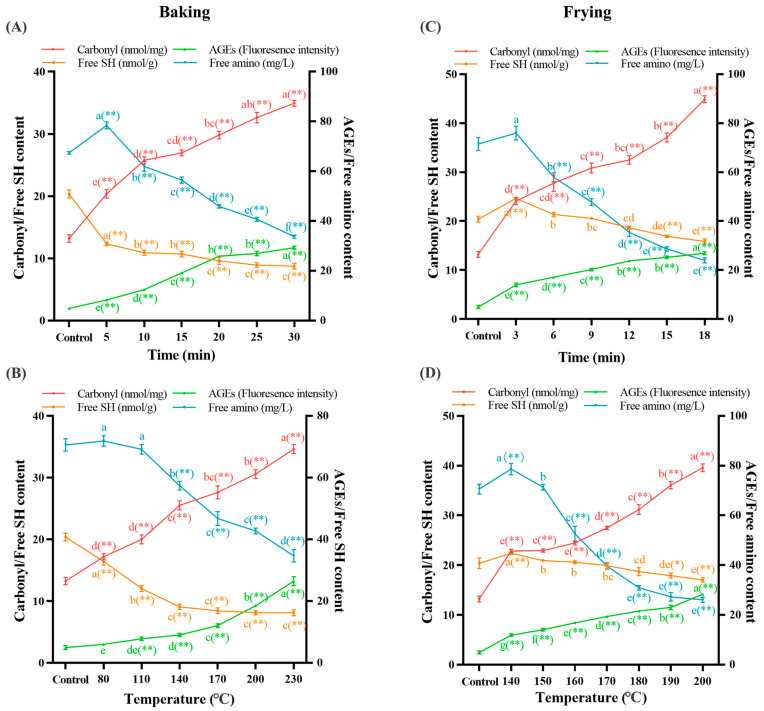
Effects of different baking times ((**A**) under 200 °C) and temperatures ((**B**) under 20 min) on dough protein carbonyl content, AGE content, free SH content, and free amino content; impacts of different frying times ((**C**) under 180 °C) and temperatures ((**D**) under 10 min) on dough protein carbonyl content, AGE content, free SH content, and free amino content. The data are displayed as mean ± SEM (*n* = 3). Different lowercase letters indicate significant differences between different baking and frying times and temperatures; the markers “*” and “**” represent the significant differences (*p* < 0.05) and highly significant differences (*p* < 0.01) vs. the control group.

**Figure 4 foods-12-04479-f004:**
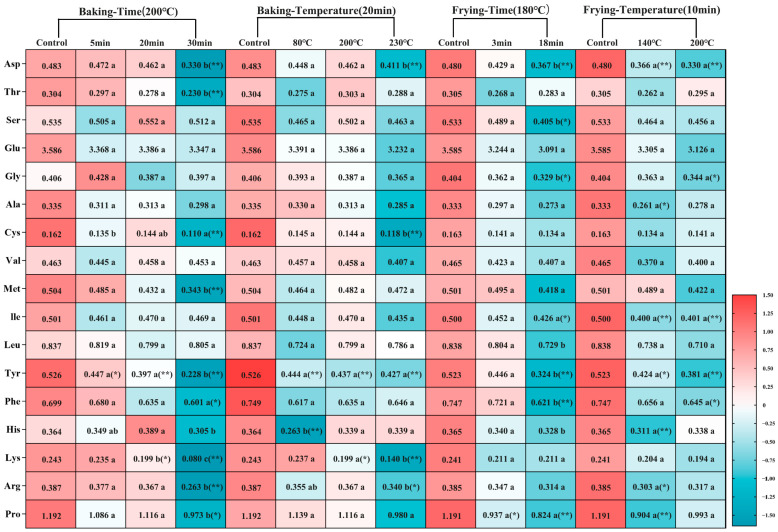
The amino acid contents were shown on the heat map (g/100 g). The heat map shows using the equation Z-score = (content in each sample − average content in all samples)/(standard deviation of value in all samples). The data are displayed as mean ± SEM (*n* = 3). Different lowercase letters indicate significant differences between different baking and frying times and temperatures; the markers “*” and “**” represent the significant differences (*p* < 0.05) and highly significant differences (*p* < 0.01) vs. the control group.

**Figure 5 foods-12-04479-f005:**
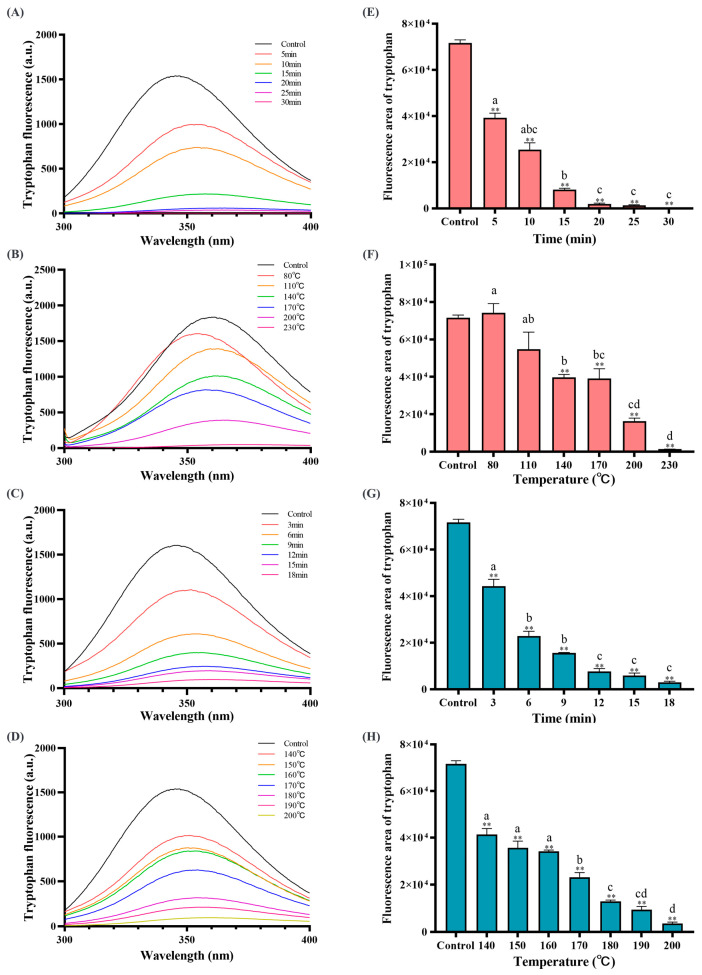
Tryptophan fluorescence spectroscopy (**A**–**D**), integrates the area of tryptophan fluorescence spectroscopy (**E**–**H**) after different baking and frying times and temperature treatments. (**A**,**E**) under 200 °C, (**B**,**F**) under 20 min, (**C**,**G**) under 180 °C, and (**D**,**H**) under 10 min. The data are displayed as mean ± SEM (*n* = 3). Different lowercase letters indicate significant differences between different baking and frying times and temperatures; the marker “**” represents the significant differences (*p* < 0.05) and highly significant differences (*p* < 0.01) vs. the control group.

**Figure 6 foods-12-04479-f006:**
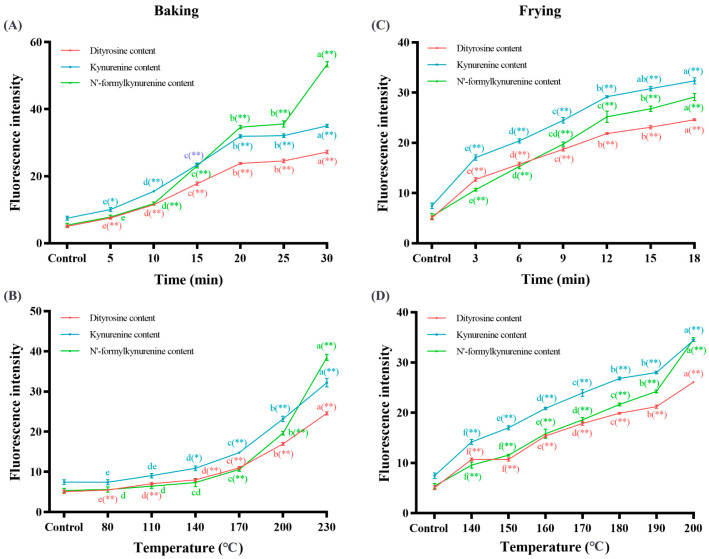
Effects of baking time ((**A**) under 200 °C) and temperature ((**B**) under 20 min) on the dityrosine content, kynurenine content, and N’-formylkynurenine content; effects of frying time ((**C**) under 180 °C) and temperature ((**D**) under 10 min) on the dityrosine content, kynurenine content, and N’-formylkynurenine content. The data are displayed as mean ± SEM (*n* = 3). Different lowercase letters indicate significant differences between different baking and frying times and temperatures; the markers “*” and “**” represent the significant differences (*p* < 0.05) and highly significant differences (*p* < 0.01) vs. the control group.

**Figure 7 foods-12-04479-f007:**
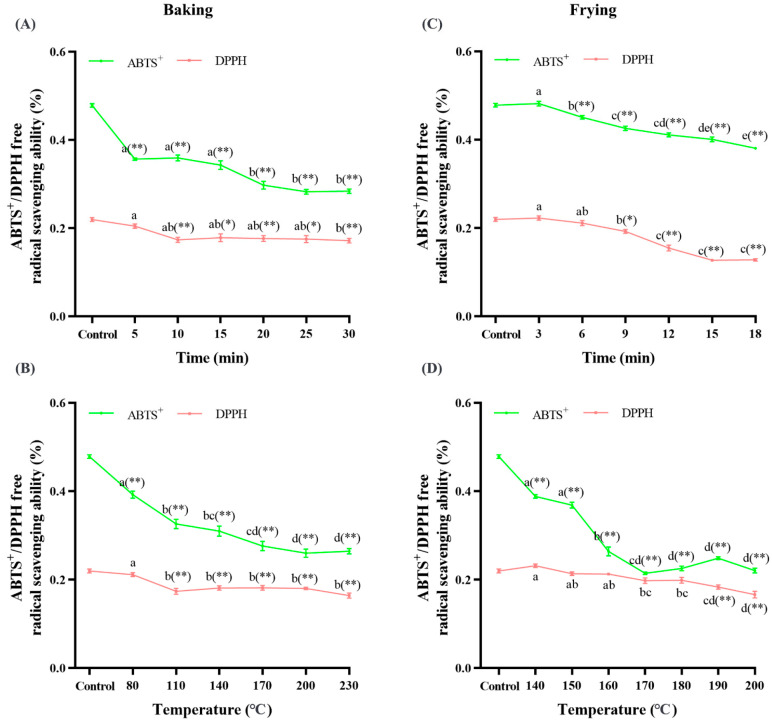
Effects of baking time ((**A**) under 200 °C) and temperature ((**B**) under 20 min) on the ABTS^+^ and DPPH free radical ability; effects of frying time ((**C**) under 180 °C) and temperature ((**D**) under 10 min) on the ABTS^+^ and DPPH free radical ability. The data are displayed as mean ± SEM (*n* = 3). Different lowercase letters indicate significant differences between different baking and frying times and temperatures; the markers “*” and “**” represent the significant differences (*p* < 0.05) and highly significant differences (*p* < 0.01) vs. the control group.

**Figure 8 foods-12-04479-f008:**
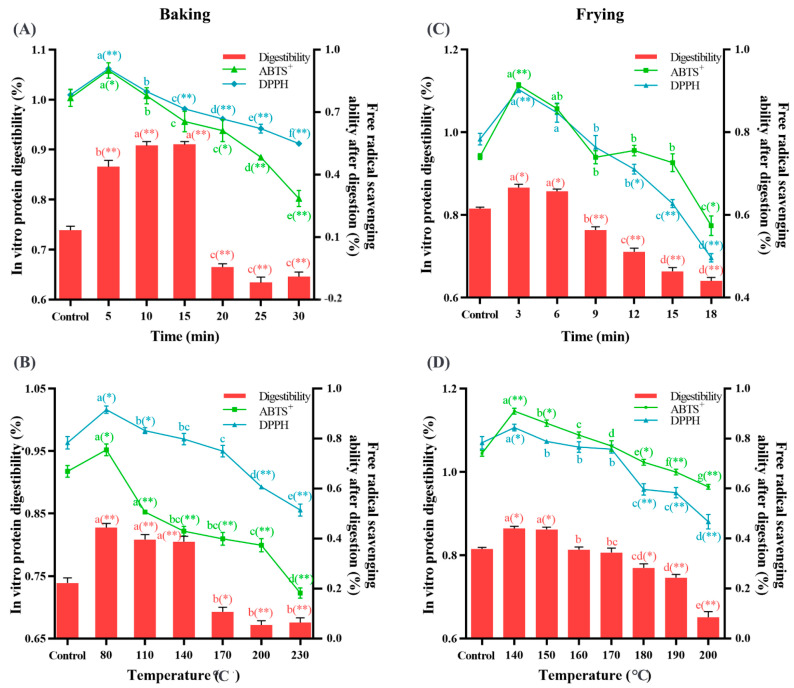
Effects of baking time ((**A**) under 200 °C) and temperature ((**B**) under 20 min) on the vitro protein digestibility and ABTS^+^ and DPPH free radical scavenging ability after digestibility; effects of frying time ((**C**) under 180 °C) and temperature ((**D**) under 10 min) on the vitro protein digestibility and ABTS^+^ and DPPH free radical scavenging ability after digestibility. The data are displayed as mean ± SEM (*n* = 3). Different lowercase letters indicate significant differences between different baking and frying times and temperatures; the markers “*” and “**” represent the significant differences (*p* < 0.05) and highly significant differences (*p* < 0.01) vs. the control group.

**Figure 9 foods-12-04479-f009:**
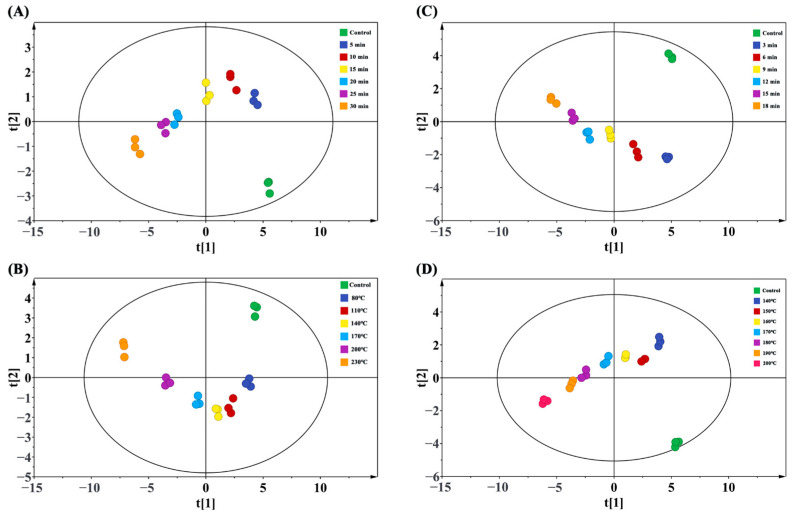
PCA score plots. (**A**) PCA (R^2^X = 0.929, Q^2^ = 0.884) in different baking times (0, 5, 10, 15, 20, 25, and 30 min under 200 °C); (**B**) PCA (R^2^X = 0.918, Q^2^ = 0.873) in different baking temperatures (80, 110, 140, 170, 200, and 230 °C under 20 min); (**C**) PCA (R^2^X = 0.925, Q^2^ = 0.867) in different frying times (0, 3, 6, 9, 12, 15, and 18 min under 180 °C); (**D**) PCA (R^2^X = 0.899, Q^2^ = 0.864) in different frying temperatures (140, 150, 160, 170, 180, 190, and 200 °C under 20 min).

## Data Availability

Data are contained within the article.

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
