# Peer review of "Effects of Baking and Frying on the Protein Oxidation of Wheat Dough"

_foods, 2023, doi:10.3390/foods12244479_

Round 1

Reviewer 1 Report

Comments and Suggestions for Authors

What is noteworthy in the assessed manuscript is the wide and comprehensive scope of research in the assessed work.

- line 90: what kind of wheat flour was used: ash , protein, dietary fiber content etc.

- throughout the work, colour identifiers are not italic fonts

- what was the column used in chapter 2.5, what was its filling?

- in chapter 2.15 not all statistical tests used in the work are provided, e.g. the Z-score test shown in Figure 4

- due to the use of methanol, is the DPPH method appropriate for wheat flour samples?

I suggest performing a PCA analysis separately for the "baking" and "frying" samples and all the marked features. Summarize all the work in a much better way.

Reviewer 2 Report

Comments and Suggestions for Authors

The study has some interesting results, but there is a lack of discussion in some areas. It would be helpful to cite more recent and relevant work and to provide more discussion of the results. Also, add experimental parameters such as the working conditions (time, temp., etc.) for baking and frying. 

The abstract and methodology section needs to be improved. Some of the methods (highlighted) are not written in a scientific way. 

Please see the attached file for detailed comments.

Comments on the Quality of English Language

Reviewer 3 Report

Comments and Suggestions for Authors

The manuscript addresses an interesting topic and is well-written.
